# Atmospheric and Ocean CO<sub>2</sub> Measurements in the South Indian Ocean Made by Two Uncrewed Surface Vehicles in 2022 and 2023

Don P. Chambers<sup>1</sup>, Jennifer Bonin<sup>1</sup>, Adrienne Sutton<sup>2</sup>, Roman Battisti<sup>2</sup>, Stacy Maenner<sup>2</sup>, Veronica Tamsitt<sup>1,3</sup>, and Nancy Williams<sup>1</sup>

<sup>1</sup> College of Marine Science, University of South Florida, St. Petersburg, FL, USA

<sup>2</sup> National Oceanographic and Atmospheric Administration, Pacific Marine Environmental Laboratory, Seattle, WA, USA

<sup>3</sup> Submarine Scientific LLC, San Francisco, CA, USA

Correspondence to: Don P. Chambers (donc@usf.edu)

Abstract. During the second half of 2022 and the first several months on 2023, a pair of Uncrewed Surface Vehicles (USVs) collected high-resolution (~5-km sampling) measurements of ocean and atmosphere pCO<sub>2</sub>, air temperature and humidity, wind, ocean skin temperature, sea surface temperature, salinity, Chlorophyll α based on fluorescence, dissolved oxygen, and ocean current velocity between roughly 13.5°E and 82°E and between the Subtropical Front (STF) and the Subantarctic Front (SAF). The mission track spanned from the Agulhas Return Current south of South Africa to the northern boundary of the Antarctic Circumpolar Current downstream of the Kerguelen Plateau. The primary goal of the mission was to collect data within cyclonic and anticyclonic eddies to quantify CO<sub>2</sub> fluxes to better understand physical processes (upwelling and downwelling) that that can contribute to carbon cycling in addition to the biological pump. In this paper, we present an overview of the mission, details on the data collected, and a preliminary look at calculated surface pCO<sub>2</sub>, separated into cyclonic/no-eddy conditions.

## 1 Introduction

20

5

The Southern Ocean south of 35°S plays a major role in the ocean carbon cycle and in Earth's climate system by accounting for ~40% of the total oceanic uptake of anthropogenic carbon dioxide (CO<sub>2</sub>) despite making up only 20% of the global ocean surface (Devries, 2014; Hauck et al, 2023). This uptake of anthropogenic carbon occurs against a background of larger natural carbon fluxes which vary both seasonally and spatially across the Southern Ocean's diverse frontal regions. Interannual and decadal variability in Southern Ocean air-sea CO<sub>2</sub> fluxes is significant (Landschützer et al., 2015; Muller et al., 2023) and drives overall variability in the global ocean carbon sink (Gruber et al., 2019).

Despite its importance in the global carbon cycle and the ongoing changes to winds, meltwater, temperature, and carbon content occurring there (e.g., Bronselaer et al., 2020; Toggweiler, 2009), the Southern Ocean is relatively under-sampled due

to its remote and dangerous nature, leading to large uncertainties in ocean carbon uptake estimates. Predictions about how the ocean carbon sink will evolve under continued anthropogenic change are foiled by our inability to fully understand it in its present state.

In the last few decades, sparse historical shipboard measurements of the ocean's partial pressure of CO<sub>2</sub> (pCO<sub>2</sub>) have been synthesized into data products such as the Surface Ocean CO<sub>2</sub> Atlas (SOCAT; Bakker et al., 2016). These products are then used along with sophisticated mapping techniques to quantify air-sea CO<sub>2</sub> fluxes and the strength of the ocean carbon sink over space and time. One such method is the self-organizing map feed-forward network (SOMFFN) of Landschützer et al. (2016). The SOMFFN incorporates sea surface temperature (SST), sea surface salinity, mixed layer depth, satellite chlorophyll, atmospheric CO<sub>2</sub>, and the gridded SOCAT pCO<sub>2</sub> product to map monthly fields of surface ocean pCO<sub>2</sub> and air-sea fluxes over time and is updated approximately annually.

40 However, the largest concentration of observations occurred in the Northern Hemisphere, with many fewer observations taken in the Southern Ocean (e.g., Fig. 3 from Bakker et al., 2016). Additionally, measurements are biased toward the summer months in the Southern Ocean due to better ocean conditions for ship-based measurements.

Autonomous platforms provide opportunities for year-round observations of important parameters of the ocean chemistry, even in the rough conditions of the Southern Ocean. The Southern Ocean Carbon and Climate Observations and Modeling (SOCCOM) Project has deployed several hundred Biogeochemical (BGC)-Argo floats throughout the Southern Ocean since 2014 (Sarmiento et al., 2023), returning thousands of profiles of temperature, salinity, O<sub>2</sub>, NO<sub>3</sub>-, and pH to depths of 2000 m approximately every 10 days. While pCO<sub>2</sub> can be derived from the BGC-Argo pH measurements and estimated total alkalinity, the measurements are expected to have a higher uncertainty than direct pCO<sup>2</sup> measurements, ~3% compared to 0.5%, along with a potential bias (Bushinsky et al., 2019; Williams et al., 2017).

50 Between January and August of 2019, an Uncrewed Surface Vehicle (USV) from Saildrone Inc. completed the first autonomous Antarctic circumnavigation (Sutton et al., 2020; 2021). It carried the NOAA PMEL-designed ASVCO2® package (Sabine et al., 2020) along with a suite of meteorological and oceanographic sensors (e.g., air and surface seawater temperature, salinity, chlorophyll fluorescence, and ocean currents), allowing highly accurate (±2 μatm, ~0.5%) measurements of surface ocean and atmospheric pCO<sub>2</sub>. Although the mission was primarily an engineering test to study the USV's endurance in the Southern Ocean's harsh conditions, data collected along the track suggested significant outward fluxes of CO<sub>2</sub> during austral winter and in regions along and south of the Polar Front where there is intense eddy activity (Fig. 2 of Sutton et al., 2021). Previous studies have theorized that mesoscale eddies could enhance or suppress the flux of CO<sub>2</sub> between the atmosphere and ocean due to the intense upwelling or downwelling caused by their circulation (McGillicuddy, 2016).

This led us to propose another Saildrone USV mission, which was funded by the National Science Foundation in late 2020. We placed an instrument package identical to that used in the 2019 Antarctic Circumnavigation experiment (Sutton et al., 2021) on a Saildrone USV (designated SD1038), which was the primary platform for our mission. We additionally leveraged collaboration with the EU-funded Southern Ocean Carbon and Heat Impact for Climate (SO-CHIC) project (https://www.sochic-h2020.eu) which was planning to operate two Saildrone USVs in the Southern Ocean around the same time. They allowed us to place a matching CO<sub>2</sub>-observing package on one of their two USV's (SD1039). Although we did not give our mission an acronym in the design or implementation phase, we have begun calling it the Southern Ocean Saildrone (SOS) mission, for reasons which will become clear shortly.

The plan was for all three SOS/SO-CHIC USVs to be launched from South Africa in the March to April 2021 timeframe. Two would sail to the SO-CHIC observation area (~10°E, 45°S), while SD1038 would sail to the SOS mission's initial observing area to the east and further south (Site A in Fig. 1), an eddy-rich area where the Subantarctic Fronts (SAF) and Polar Fronts (PF) often merge. The plan was for SD1038 to take observations of pCO<sub>2</sub>, pH, and other parameters within a number of cyclonic and anticyclonic eddies beginning in May to June. Eddies would be identified in near-real-time maps of sea surface height anomalies observed by satellite altimeters (e.g., Chelton et al., 2007; Mason et al., 2014). After observing as many eddies as possible in the two-month window, SD1038 would transit along the Polar Front until it reached the eddyrich region downstream of the Kerguelen Plateau (Site B in Fig. 1). SD1039, after making observations in the SO-CHIC region, was expected to follow SD1038 with a lag of 3-4 weeks, trying to sample the same eddies SD1038 had previously.

Figure 1: Trajectories of SD1038 (orange diamonds) and SD1039 (blue and white circles) on top of standard deviation of sea surface height variability (color contours). SSH variability calculated from gridded multi-mission sea level anomaly maps distributed by Copernicus (<a href="https://doi.org/10.48670/moi-00148">https://doi.org/10.48670/moi-00148</a>). Approximate positions of the Subtropical Front (STF), Subantarctic Front (SAF) and Polar Front (PF) from Orsi et al. (1995) shown with black dots. The original planned trajectory and

observation sites are shown in orange arrows and circles. Contours are bathymetry at 1000m intervals, and the Kerguelen Plateau (KP) is highlighted. The white X indicates the location of Crozet Island, where atmospheric CO<sub>2</sub> measurements are collected routinely.

85

Early on, it became clear that the SOS mission plan would have to be altered. Due to closures of South Africa due to the COVID-19 pandemic, both the SOS and SO-CHIC missions were delayed by more than a year. SD1038 finally reached a site west of the initial observation area on July 19, 2022, many months in the season later than expected. However, it quickly became apparent that the wave-generator and solar cells were not recharging the batteries fast enough to keep up with the requirements from the instrumentation and navigation, so the USV was rapidly losing power. On July 23, a lower sampling rate was implemented to reduce power drain with a hope that the batteries could recharge. But by July 26, it was clear the hydrogenerator couldn't operate in the high sea state, and that the rudder or wing had been damaged (possibly by a rogue wave), limiting the ability to tack to the port side. Consequently, the USV was drifting south. All instruments were turned off to conserve power in an attempt to recover the drone, allowing only a short period of time for observations (Fig. 1).

95

100

Subsequently, both the SOS and SO-CHIC missions were reassessed. The two USVs for the SO-CHIC mission had departed South Africa significantly later than SD1038 and had just crossed the Agulhas Current when SD1038 was powered down to try to limp it back to port in Johannesburg. After numerous discussions between the SOS Mission team, the SO-CHIC principal investigator, and the Saildrone navigation team, it was decided that SD1039 would begin moving toward the eddyrich area downstream of the Kerguelen Plateau, but at a latitude no further south than 45°S for several months until increasing summertime solar radiation became sufficient to keep the batteries charged via the solar panels. The "new" SOS mission began on September 1 when SD1039 began moving eastward around 42°S, 12°E (Fig. 1). In the meantime, all contact had been lost with SD1038. Its last known position was 51°S, 24°E. However, not all instruments were powered on immediately due to ongoing power management issues, so many measurements are missing in this early period until about Sept. 16. In particular, measurements from the ASVCO2® system are not available until Sept. 16.

110


The "new" SOS mission plan was to observe eddies of opportunity along the Subtropical Front (STF) while moving eastward as quickly as possible toward the second main observation area downstream of Kerguelen between the Subantarctic Front (SAF) and Polar Front (PF), where it was hoped the USV could sample individual eddies over a slightly longer period. SD1039 reached a region just north of the proposed sampling area (B in Fig. 1) in early January 2023, but by the end of the month (January 26) it was clear that it was becoming more difficult to navigate the USV and that complex maneuvering (i.e.: targeting of specific eddies and intentionally sampling in patterns inside them) in the high sea-state was not possible. Efforts were made to steer the drone to Australia for recovery. Unfortunately, it became clear that the drone would not be recovered (like SD1038), and so the majority of the instruments were turned off on February 27, 2023.


Fortunately, many of the primary observations had been uploaded in near-real-time throughout the mission, primarily to be used for measurement assessment and to aid in directing the USV, so we are able to report and archive the primary science measurements of both SD1038 and SD1039. Interestingly, the ASVCO<sub>2</sub> system (Sabine et al., 2020) continued to operate after most of the other instruments were turned off, until April 27, 2023, presumably because Saildrone, Inc. could not power it down or its independent battery system maintained sufficient charge to take observations and transmit them. Therefore, hourly data for some variables (including positions from an onboard GPS receiver) continued to be shared with the data server at the NOAA Pacific Marine Environmental Laboratory (PMEL).

While the amount of data, locations, and timing of CO<sub>2</sub> measurements made from the two USVs as part of the SOS mission were not as anticipated, they still represent important direct observations of the carbon system in a poorly sampled region of the ocean. In this paper, we will describe the principal results of the mission, including how we deduced eddy matchups. Section 2 will provide an overview of the instruments onboard and the data collected, the methodology for determining eddy matchups, and describe where the data are permanently archived and how they can be accessed. Section 3 will discuss results from the primary CO<sub>2</sub> measurement system, showing derived atmospheric and ocean pCO<sub>2</sub> values along the tracks, discuss when the track was in an eddy, and provide some brief analysis of the results within eddies compared to when the USVs were not in an eddy. However, a thorough scientific analysis is beyond the scope of this paper and is left for further studies. This document is primarily intended to provide an overview of the data and mission.

#### 2 Instrumentation and Data Collected

The Saildrone Uncrewed Surface Vehicle (USV) is an autonomous ocean data collection platform designed for long range, long duration missions of up to 12 months. Saildrone USVs run solely on renewable energy, using wind power for propulsion and solar energy and wave generators to run a suite of science-grade sensors. Each vehicle consists of a 7 m narrow hull, a maneuverable wing for sailing, and a keel with a 2.5 m draft. The USV weighs approximately 750 kg and can be launched and recovered from a dock. The USVs used in this mission are modified versions for the one used in the 2019 circumnavigation of Antarctica (Sutton et al., 2021). Anyone interested in specifics on Saildrone USVs should refer to that paper and all relevant references within it. The only major change to the USVs used for the SOS mission was a shorter and hardened wing designed to accommodate higher waves and winds in our mission area (Ricciardulli et al., 2022; Chiodi et al., 2024).

There is a suite of science grade sensors on each Saildrone platform to measure key atmospheric and oceanographic environmental variables (Table 1). These include solar irradiance, longwave radiation, atmospheric pressure, air temperature

and humidity, wind speed and direction, ocean skin temperature, sea surface temperature (SST), salinity, Chlorophyll  $\alpha$  based on fluorescence, and dissolved oxygen, among others (Zhang et al., 2019). However, it must be noted that due to ongoing power issues, only the seawater properties, the ASVCO2® system, and the ADCP system were operated nearly continuously with minimal outages. Most other parameters have significant gaps due to instruments being turned off to conserve power, as they were deemed of lesser importance compared to maintaining the USV power and primary instruments. We have archived all available observations in the databases (Table 2), but users should not expect complete records of many variables, other than seawater properties and chemistry data, The basic atmospheric and oceanographic data were sampled at hourly or sub-hourly intervals for SD1038's entire record and for 1 September 2022 through 27 February 2023 for SD1039 (the cyan portion of the trajectory shown in Fig. 1), after which point most of the sampling systems were turned off to conserve power.


Table 1: Primary measurements on SD1038 and SD1039, including instrument type and special notes on placement or availability.

| Measurements<br>(incl. Variable Names in<br>Datafiles)                         | Instrument                                                     | Notes                                                     |
|--------------------------------------------------------------------------------|----------------------------------------------------------------|-----------------------------------------------------------|
| wind parameters (WIND_U, WIND_V, WIND_W, WIND_SPEED, WIND_GUST, WIND_FROM_DIR) | Gill model 1590-PK-020<br>anemometer                           | Wind values measured at ~3.4 meters above local sea level |
| Satellite wind speed (CCMP_WIND_EAST, CCMP_WIND_NORTH, CCMP_WIND_SPEED)        | Cross-Calibrated Multi-Platform (CCMP) Wind Vector analysis    | Interpolated from 0.25°, 6-hour grids.                    |
| Atmospheric<br>temperature/humidity<br>(ATM_TEMP,<br>ATM_REL_HUMID)            | Rotronic model HC2-S3 standard meteorological probe            | Install height: 2.3 m                                     |
| Photosynthetically Active Radiation (PAR)                                      | LI-COR model LI-192SA<br>underwater quantum sensor             |                                                           |
| Incoming Shortwave Radiation<br>(IRRAD_SW_DIFFUSE,<br>IRRAD_SW_TOTAL)          | Delta-T model SPN1-shaded shaded pyranometer                   | Only on SD1039                                            |
| Outgoing longwave radiation<br>(IRRAD_LW)                                      | Eppley model PIR infrared radiometer                           | Only on SD1039                                            |
| Atmospheric Pressure (ATM PRESS)                                               | Valsala model PTB210 barometer                                 | Install height: 0.2 m                                     |
| Seawater properties (SW_COND, SW_TEMP,                                         | Sea-Bird Scientific model SBE37-SMP-ODO microCAT conductivity, | Install height: -0.5 m                                    |

|                             | · · ·                                      | T                                  |  |
|-----------------------------|--------------------------------------------|------------------------------------|--|
| SW_SAL, O2_SAT, O2_CONC)    | temperature, and optical pressure          |                                    |  |
|                             | recorder with dissolved oxygen             |                                    |  |
|                             | sensor                                     |                                    |  |
| Skin temperature            | Heitronics model CT15.10 infrared          | Install height: 2.3 m              |  |
| (SW_TEMP_SURFACE_SKIN)      | radiation thermometer                      |                                    |  |
| Chlorophyll                 | WET Labs model FLS fluorometer             | Install height: -0.5 m             |  |
| (CHLOR)                     |                                            |                                    |  |
| Longitude/Latitude, wave    | VectorNav model VN-300 GNSS-               |                                    |  |
| characteristics             | aided inertial navigation system           |                                    |  |
| (WAVE DOM PERIOD,           |                                            |                                    |  |
| WAVE SIG HEIGHT)            |                                            |                                    |  |
| Water velocity              | Teledyne model Workhorse                   | Install height: -1.9 m             |  |
| (VEL EAST, VEL NORTH,       | WHM300-I-UG1 acoustic doppler              | SD1038: never turned on            |  |
| VEL UP)                     | current profiler (ADCP)                    | SD1039: intermittently on          |  |
| _ /                         |                                            | during mission                     |  |
| Atmospheric/Ocean Chemistry | ASVCO <sub>2</sub> with Licor model LI-820 | Install heights: 1.3 m (air) and - |  |
| (ATM fCO2, ATM H2O,         | gas analyzer                               | 0.5 m (ocean)                      |  |
| ATM pCO2,                   |                                            | Included separate GNSS system      |  |
| ATM PRESS LICOR,            |                                            | for recording positions            |  |
| ATM TEMP LICOR,             |                                            |                                    |  |
| ATM xCO2 DRY,               |                                            |                                    |  |
| ATM xCO2 WET, O2 RATIO,     |                                            |                                    |  |
| DIFF fCO2, DIFF pCO2,       |                                            |                                    |  |
| SW_fCO2, SW_H2O, SW_pCO2,   |                                            |                                    |  |
| SW xCO2 DRY,                |                                            |                                    |  |
| SW xCO2 WET)                |                                            |                                    |  |

The vehicles used for the SOS mission were also equipped with acoustic doppler current profilers (ADCP). However, the ADCP was never turned on for SD1038 due to power consumption problems and was on intermittently for SD1039 due to several issues. No ADCP data was collected before 26 September 2022, or from 18 October 2022 1530 UTC to 26 October 2022 0 UTC. Additionally, between 26 September 2022 1700 UTC and 12 October 2022 2100 UTC, the ADCP data is flagged as "bad" within the datafile and should be used with caution, because the ADCP was accidentally switched to bottom tracking mode in deep water during this period.


Although both USVs had an anemometer to directly measure wind conditions, the system failed early in the SD1039 leg (on 2 September 2022). Because of this, we have also included windspeed computed from a statistical combination of satellite-based vector winds and atmospheric re-analyses, the Cross-Calibrated Multi-Platform (CCMP) Wind Vector analysis product (<a href="https://www.remss.com/measurements/ccmp/">https://www.remss.com/measurements/ccmp/</a>; Mears et al., 2022a,b). These data, collocated at USV times and locations, are included even when the anemometer winds are available so users can have a consistent wind data set and can compare in situ and satellite-based wind speed.

The primary instrument package for this mission was the ASVCO<sub>2</sub> system (Sabine et al., 2020), identical to the system deployed on the 2019 Saildrone mission (Sutton et al., 2021). The ASVCO<sub>2</sub> system is capable of measuring surface ocean and atmosphere pCO<sub>2</sub> to within  $\pm$  2  $\mu$ atm ( $\pm$  0.5%) by performing a calibration before every measurement with a zero and an on-board CO<sub>2</sub> gas standard and have been used on over a dozen missions. For anyone interested in details on how the system works and the exact processing steps to convert between measured variables and derived variables, we refer you to Sutton et al. (2014) and Sabine et al. (2020).

To be consistent with the ASVCO<sub>2</sub> sensor data distributed with the 2019 Saildrone mission, the SD1038 and SD1039 ASVCO<sub>2</sub> data are archived at the NOAA National Centers for Environmental Information (NCEI) (Chambers et al., 2025a,b) in the same format and with the same processing as done for previous USV-based surface ocean pCO<sub>2</sub> data. The raw wet xCO<sub>2</sub> data, temperature, salinity, and pressures are included so other data users can recalculate dry xCO<sub>2</sub>, fCO<sub>2</sub>, and pCO<sub>2</sub>. While the ASVCO<sub>2</sub> sensor package also included a DuraFET pH sensor, these data are not included in the files as they are uncalibrated. They were only used (along with internal CO<sub>2</sub> system diagnostics) to quality check and flag CO<sub>2</sub> measurements. This was done by calculating covariance of pH and CO<sub>2</sub> observations over segments where there were significant CO<sub>2</sub> deviations. Existence (or lack) of covariance between CO<sub>2</sub> and pH outliers was used as independent evidence that CO<sub>2</sub> data were good (or questionable).

The two ASVCO<sub>2</sub> datasets do not contain all the ancillary data measured by other Saildrone USV instruments, nor do they contain any sub-hourly observations, since the system was not linked to the transfer system used to download the other observations via satellite link. Because of this, we have created a third dataset that includes all the ASVCO<sub>2</sub> variables in the two NCEI files as well as all other available observations from each of the two USVs (Chambers et al., 2025c). The only exception is the segment of SD1039 shown in white in Fig. 1. These data come from only the ASVCO<sub>2</sub> system; other sampling systems had been turned off to conserve power at this point. The measurements relevant to the chemistry data for that short leg are available in the NCEI archive (Chambers et al., 2025a), including positions recorded by a GPS internal to the ASVCO<sub>2</sub>. Because there was not a full suite of measurements, including such key parameters as ocean temperature and salinity, as in the primary mission (cyan track in Figure 1), this later CO<sub>2</sub>-related data are not included in the full mission datafile for SD1039. The details of the datafiles (including archive location) are given in Table 2.

Table 2: Archived Data Files (including locations and important differences)





| Datafile              | Location                           | Notes                                   |                               |
|-----------------------|------------------------------------|-----------------------------------------|-------------------------------|
| SD1038 Chemistry Data | https://doi.org/10.25921/r2mt-t398 | Measured and parameters from the system | derived<br>ASVCO <sub>2</sub> |
| SD1039 Chemistry Data | https://doi.org/10.25921/6b0k-r665 | Measured and                            | derived                       |

|                               |                                       | parameters from the ASVCO <sub>2</sub> system |
|-------------------------------|---------------------------------------|-----------------------------------------------|
| SD1038/1039 Full Mission Data | https://doi.org/10.17632/9ymsjsyhhp.1 | Chemistry and other physical                  |
|                               |                                       | measurements not in the                       |
|                               |                                       | Chemistry Data files. Hourly                  |
|                               |                                       | data for all, sub-hourly for some.            |

For users familiar with the Surface Ocean CO2 Atlas (SOCAT, <a href="https://socat.info">https://socat.info</a>), the chemistry data for SD1039 has been uploaded and is currently available in the current database (SOCAT-v2025, Bakker et al, 2025). The data for SD1038 had not been processed in time for inclusion in the current atlas but will be added in the next one.

The observables listed in Table 1 were sampled at various rates. For example, the ASVCO<sub>2</sub> returned a measurement based on a several minute average at 16 minutes after each hour. Other observations made using the primary Saildrone USV instrumentation packages were transmitted on more frequent intervals, some as frequently as every minute, some 10-15 times each hour. Each reported sample is the average of ~11 observations taken in a planned burst, over an 11-second span centered at the top of the minute reported. To account for the different sampling rates, we have created two types of data files for users in the full mission data set (Table 2): one with one-minute sampling, the other with one-hour averages. The one-minute files have many missing records due to the non-constant sampling (e.g., for the chemical variables, only one per hour). Where available, the standard deviation of the 11-second burst measurement is also included, which can be used as an estimate of the precision of the measurement. Because the majority of data types do not vary rapidly within an hour, we additionally made a smaller hourly-averaged files, based on averages of the sub-hour sampled files. The standard deviation of the minute-sampled data can also be used as a measure of the accuracy of the 1-hour averages. For the air and seawater CO<sub>2</sub> variables where there was only one observation within that hour only that single value is given, so it is not a true hourly average.

Because the goal of the SOS mission was to measure pCO<sub>2</sub> within different eddies, we have also provided an estimate of whether the USV was in an eddy or not, along with the type of eddy (cyclonic, anticyclonic) in the main mission datafile. The variable EDDY\_DIRECTION in the file has three values (1 for anticyclonic, -1 for anticyclonic, and 0 for not within an eddy). This was not done for SD1038 as the drone lost power before we could intentionally maneuver it into any eddies, but has been done for SD1039. Eddy direction is determined using a database of eddy positions and sizes that are provided in the regularly updated near-real-time Mesoscale Eddy Trajectory Atlas (META, 2025), which is distributed by the E.U. Copernicus Marine Service/CNES/CLS and is based on satellite altimetry estimates of absolute dynamic topography using the detection methods of Mason et al. (2014). The eddy database contains all the statistics of the eddy necessary for this study: the time, location of the center, the amplitude of sea level anomaly in the center, radius of the eddy from the center to where the velocity is the maximum ( $r_{max}$ ), and the maximum velocity.

We describe SD1039 as being "inside" an eddy if the distance between SD1039 and the reported center of the eddy is less than the reported radius of that eddy on the same day that the USV passed through it. Additionally, we require the USV to remain within the radius of the (moving) eddy for at least 24 hours. The latter constraint was required to prevent four very short (2-17 hour) eddy "intersections" which occurred as the distance from the eddy center to SD1039 approached the eddy radius. Investigations using SD1039's ADCP data did not support the presence of an eddy in these cases, so they were rejected. Additionally, in the case of the first cyclonic eddy, it appeared that SD1039 moved into the eddy, out of it, and then back in. Upon investigation, this odd behavior was caused by an atypically large shift of the eddy's position between one day's database record and the next, which is unlikely to be realistic. In our product, we thus define all times between the initial entrance and final exit of that eddy to be "in an eddy", even when the database says it is slightly outside the radius of the eddy. Altogether, SD1039 traveled through 12 eddies as defined by these criteria, of which 9 were anticyclonic and 4 were cyclonic (Fig. 2). However, only 8 of the anticyclonic eddies have CO<sub>2</sub> measurements – the first eddy at the start of the transit has no valid ASVCO<sub>2</sub> measurements.

Figure 2: Trajectory of SD1039 with nearby eddies as found via matchups with an altimeter-based eddy atlas. Track colors represent periods when SD1039 was in a region of no eddies (white), an anticyclonic eddy (red), or a cyclonic eddy (blue). Center locations of expected eddies from the database, averaged over the period SD1039 is nearby, are shown as dots, with dot color representing average database amplitude and black circles denoting the average radius of each eddy. Eddies are numbered based on type (cyclonic=C, anticyclonic=AC) along the track where pCO<sub>2</sub> measurements are available (discussed in Section 3).

Figure 3: Surface ADCP velocities (vectors) from SD1039, along with the SD1039 track colored in the same manner as in Figure 2. Red colors indicate a nearby expected anti-cyclonic eddy, while blue colors indicate a nearby cyclonic eddy. Colored circles show the expected mean center of eddy rotation and its amplitude, averaged over the time SD1039 was inside it, based on the eddy database.

The eddy database is capable of detecting only large eddies (diameters > 100-200 km), due to the use of gridded, optimally interpolated altimetry data in its construction. We anticipate that SD1039 likely passed through additional smaller eddies on its path. We attempted to use the ADCP data to detect rotations associated with such eddies but found that impossible using a single track of ADCP data. However, with additional work (e.g., removing some climatological currents to obtain anomalous velocity and removing non-geostrophic Ekman currents using wind fields), more eddy-related information might be teased out of this ADCP data. Therefore, we include the ADCP velocity information in the data files for users to experiment with.

#### 3 Discussion of Observations






Here we only analyze observations from the SOS mission and present a preliminary analysis of observed variations that may correlate with the eddy-type. Observations cannot be directly compared to other measurements made by ships, drifters, or other USVs because no other observations were made within 500 km or within 3 months of the two USV transits (based on the current SOCAT database). While one goal of the mission was to attempt a crossover with a biogeochemical Argo float (as done with the 2019 Saildrone mission (Sutton et al., 2021)), this was not possible during the SOS mission. Any observations made near the same area were collected perhaps a year earlier or later. The 2019 Saildrone mission, for example, occurred 3-4 years earlier and crossed the South Indian Ocean farther south than SD1039. Any comparison would have to account for spatial and temporal differences and is best suited for a scientific investigation which is beyond the scope

of this data description paper.






Atmospheric pCO<sub>2</sub> has a mean value of 410 μatm with a standard deviation of 3.5 μatm (Fig. 4). Observations of atmospheric CO<sub>2</sub> made at Crozet Island in the Indian Ocean (46.4°S, 51.8°E) between Sept. 4, 2022 and April 26, 2023 has a mean of 415.2 ppm (standard deviation = 0.46). Converting to pCO<sub>2</sub> in μatm using using average and fixed air pressure (1 atm) and water vapor pressure (0.015 atm), this corresponds to approximately 409 μatm. Recalling that the accuracy of pCO<sub>2</sub> measurements from the ASVCO<sub>2</sub> system is ± 2 μatm, the USV measurements are consistent with observed pCO<sub>2</sub> in the region. The Crozet CO<sub>2</sub> measurements were downloaded from the NOAA Global Monitoring Laboratory (https://gml.noaa.gov/data/dataset.php?item=crz-co2-flask; Lan et al., 2025) on 30 Jul 2025.

The most obvious signal in measured oceanic pCO<sub>2</sub> by SD1038 during its June-July transit from South Africa is an increase in pCO<sub>2</sub> values from ~350 μatm at 35° S to ~405 μatm at 50° S as the vehicle moved southwards (Fig. 4a and 4b). These values are within an expected range, as Shadwick et al. (2023) documented seasonal variations at a similar latitude in a mooring south of Tasmania with a peak (380-400 µatm) around July/August. Comparing the pCO<sub>2</sub> to a mean monthly climatology (Fig. 5) we confirm the shift in values of oceanic pCO<sub>2</sub> in SD1038 around 45°S is consistent with the mean state in that region for the time of year. We do note a bias between the measurements of both SD1038 and SD1039 and the climatology (Fig. 5) of approximately 25-30 µatm (SD1038/1039 higher). The climatology was based on an average of observations from 1998 to 2015. Although the exact epoch represented by the climatology is not explicitly given in the reference paper or dataset (Landschützer et al., 2020a,b), P. Landschützer confirmed to us via email that the mean epoch is 2006-2007. The mean rate of change in atmospheric pCO<sub>2</sub> at Crozet since 2005 is ~ 2 µatm yr<sup>1</sup> (based on the approximate pressure values stated previously). Multiplying this rate by the time difference (15-16 years) between our observations and the climatology epoch gives a climate-induced change of 30 to 32 µatm. Since oceanic pCO<sub>2</sub> should follow trends in atmospheric pCO<sub>2</sub> assuming an equilibrium state (e.g., Fay et al., 2024), one would expect measurements of oceanic pCO<sub>2</sub> to have changed by this much on average. This is approximately the bias we observe, so we conclude the observed bias with the climatology is primarily due to increasing CO<sub>2</sub> concentrations since 2006-2007 and any smaller deviations are interannual fluctuations and using direct pressure/temperature observations instead of climatological means.

Figure 4: a) Observed ocean pCO<sub>2</sub> along the transects of SD1038 and SD1039. Also shown versus time for SD1038 (b) and SD1039 (c), along with atmospheric pCO<sub>2</sub> (black dots). Mean front positions from Orsi et al (1995) are also shown as thin black lines in a).


Figure 5: Observed ocean pCO<sub>2</sub> along the transects of SD1038 (a) and SD1039 (b). Also shown are values from a mean monthly climatology (red, data from Landschützer et al., 2020a,b). The climatology is based on all available data from 1988 to 2015 and so will reflect a mean state in 2006-2007 (P. Landschützer, personal communication).

The ocean pCO<sub>2</sub> along the SD1039 track (Fig. 4a,c) primarily varies between 360 - 385 µatm, except for short excursions where the water pCO<sub>2</sub> abruptly drops or rises by up to 20 µatm for a period lasting less than a day (in most cases, only a few hours). Generally, the ocean pCO<sub>2</sub> is lower than the atmospheric pCO<sub>2</sub>, indicating the ocean was acting as a sink for CO<sub>2</sub>

during the USV transits. Near the end of the SD1038 transit (south of 45°S), ocean pCO<sub>2</sub> increases to be close to that of the atmospheric pCO<sub>2</sub>. Some spikes are larger than +40 µatm near the end of the record for SD1039, indicating a short-term higher concentration of pCO<sub>2</sub> in the surface waters than in the atmosphere. This suggests the potential for outgassing of CO<sub>2</sub> from the ocean to the atmosphere during these periods, such as was observed in the 2019 Saildrone mission (Sutton et al., 2021), but more work would be required to fully quantify this.

The extended drop in ocean pCO<sub>2</sub> in October 2022, when values reached as low as 340 μatm occurred around latitude 40° S and between 37° E and 47° E. At the same time, there was a steady rise in Chlorophyll α (Chl), rising to the maximum values observed during the transect (> 20 mg/l, averaged over 2 days) (Fig. 6). The time of transit was austral spring, when phytoplankton blooms tend to be frequent (e.g., Bathmann et al., 1997). While the minimum pCO<sub>2</sub> does not occur at the same time as maximum Chl, we also have no direct measure of the age or evolution of the possible phytoplankton bloom in the area.

We note that the SST dropped by nearly 5°C during this period (Figure 7), suggesting increased upwelling during this period. This is consistent with increased nutrient availability for a phytoplankton bloom. The temperature drop is unlikely to be the primary reason for the pCO<sub>2</sub> change, as a shift in SST from  $\sim$ 16°C (just before the drop) to 11°C in the period of low pCO<sub>2</sub> (Figure 7) will only cause a change of change of  $\sim$  1  $\mu$ atm, which is 10% of the observed change (Figure 6).

Because SD1039 transited near the middle of the austral spring, it is possible that the drop in ocean pCO<sub>2</sub> is related to a previous spike in Chlorophyll  $\alpha$  (phytoplankton) concentration. We note this as a potential area of interest for future studies, as it might be possible to derive a time-series of Chlorophyll  $\alpha$  for this area using satellite ocean color observations. That is, however beyond the scope of this study.


Figure 6: Low-pass filtered ocean pCO<sub>2</sub> (blue, right scale) and Chlorophyll  $\alpha$  (red, left scale) along SD1039 transect. The low-pass filter was a Gaussian smoother with a roll-off of 24 hours, which effectively suppresses variations with periods shorter than 2-days.

Figure 7: a) Observed SST along the transects of SD1038 and SD1039. Also shown versus time for SD1038 (b) and SD1039 (c), along with upper 5m temperature from monthly gridded Argo maps (Roemmich and Gilson, 2009), designated SIO for Scripps Institute of Oceanography. Mean front positions from Orsi et al (1995) are also shown as thin black lines in a).

The surface salinity (SSS) is also plotted (Figure 8). It is apparent from both Figure 7 and 8 the changes in SST and SSS as the two USVs cross fronts. This is particularly noticeable in the measurements of SD1038 as it crosses the STF, the SAF, and the PF. It is seen to a lesser extent when SD1039 moves into the Subantarctic Zone between the STF and SAF in December 2022. Note that the maximum gradients in SST and SSS do not perfectly align with the mean frontal positions of Orsi et al (1995). Fronts are highly variable in time and space, and mean frontal positions are only an approximation. For example, Kosempa and Chambers (2014) noted that a frontal calculation based on upper ocean zonal transport placed the STF south of Africa substantially further north than in Orsi et al. (1995), which is also observed by SD1038 in SST (Figure 7) and SSS (Figure 8).


Figure 8: a) Observed SSS along the transects of SD1038 and SD1039. Also shown versus time for SD1038 (b) and SD1039 (c), along with upper 5m temperature from monthly gridded Argo maps (Roemmich and Gilson, 2009), designated SIO for Scripps Institute of Oceanography. Mean front positions from Orsi et al (1995) are also shown as thin black lines in a).

We considered whether there was any relationship between abrupt changes in ocean pCO<sub>2</sub> and SD1039's location within an eddy (Fig. 9) but found no correlation between the magnitude of change or direction with being in an eddy or the type of eddy, even near the end of the transect when the variations are larger. For example, while there is a significant increase in ocean pCO<sub>2</sub> during the transect of anticyclonic eddy 6 (AC6), during other transects (AC2, 3, 7, 8) there are either no major changes, or even decreases in pCO<sub>2</sub>.

The low number of cyclonic eddy transects does not allow any robust statistical comparisons, but qualitatively we find no consistent pattern. While there is a small increase in pCO<sub>2</sub> in cyclonic eddy 2 (C2) up from the minimum values likely connected with a phytoplankton bloom, this does not occur in C1. In the long transit of C3, there are rises and falls of pCO<sub>2</sub>. The largest multi-day increase in pCO<sub>2</sub> occurs outside an eddy in early 2003 (just before entering AC8).

While these results are not conclusive that upwelling/downwelling in eddies has no effect on ocean pCO<sub>2</sub>, it is apparent it is at most a second order effect compared to other processes along the USV transit during the austral spring and summer. This is not surprising, as biological interactions are strong during this period.

Figure 9: Ocean pCO<sub>2</sub> (blue) along SD1039 transect and eddy flag values (red). Flags with value =1 indicate an anticyclonic eddy, while value=-1 indicates the drone was in a cyclonic eddy.

## 4 Data Availability




All data are available from public archives. The observations (and derived chemistry variables) from the two ASVCO<sub>2</sub> system are stored in two files, separated by the designation of the USV: SD1039 (https://doi.org/10.25921/6b0k-r665; Chambers et al., 2025a) and SD1038 (https://doi.org/10.25921/r2mt-t398; Chambers et al., 2025b). These data are in CSV format (identical to previously released ASVCO<sub>2</sub> data from a previous (2019) USV mission (Sutton et al., 2020). A third set of files (in netCDF format) includes both datasets as well as additional ocean and atmospheric observations (e.g., those not listed as Atmospheric/Ocean Chemistry in Table 1): https://doi.org/10.17632/9ymsjsyhhp.1; Chambers et al., 2025c). This combined, full mission dataset also includes hourly averages and sub-hourly data (when available) and flags for whether the USV was in an eddy and the type of eddy.

### **5 Conclusions**

While the original goals and design of the SOS mission had to change dramatically due to outside circumstances, the team modified the mission goals to obtain as many useable atmospheric and oceanic observations as possible, in regions of the south Indian Ocean that are rarely sampled. Instead of simply navigating SD1039 back to South Africa after the problems encountered with SD1038, we attempted to navigate SD1039 to an eddy-rich area downstream of the Kerguelen Plateau between the Polar and Subantarctic Fronts, albeit along a more northerly route than planned and during the austral spring and summer, not winter.

SD1039 did transit through a handful of cyclonic and anti-cyclonic eddies, collecting novel CO<sub>2</sub> measurements. Although we were not successful in sampling eddies in a systematic manner (or with any time delay using two USVs), these measurements will contribute to a growing database of such data within eddies in the Southern Ocean (e.g., Keppler et al., 2025). The SOS team hopes that the small (but high-resolution) observations collected by SD1038 and SD1039 during the mission will aid future investigations in better understanding the physical processes that help control carbon cycling in the Southern Ocean.

#### **Author Contributions**

DPC managed day-to-day science operations and collection during the missions and supervised analysis and archiving post-mission. He also wrote much of this data description document. JB provided analysis of eddies during the mission and handled the post-mission processing of the non-ASVCO<sub>2</sub> data. She also helped in producing figures for the manuscript and in the eddy-matchup analysis post-mission and also co-wrote Sect. 2. AS, SM, and RB performed post-mission processing and quality control of the ASVCO<sub>2</sub> data and added commentary on the analysis of the results. VT aided in underway science operations, in the eddy match-up algorithms, and in writing the Introduction material. NW conceived the mission, wrote the original proposal, and co-wrote the Introduction material with DPC and AS.

## Acknowledgements



We would like to thank the engineers at Saildrone, Inc. for managing the Saildrone USVs under challenging conditions and downloading as much data as possible when it was clear SD1039 would not be recovered (in particular, Julia Paxton and Matt Womble). The altimetric Mesoscale Eddy Trajectories Atlas (META3.2exp NRT) is produced by SSALTO/DUACS and distributed by AVISO+(https://www.aviso.altimetry.fr/en/data/products/value-added-products/global-mesoscale-eddy-trajectory-product/meta3-2-exp-nrt.html) with support from CNES, in collaboration with IMEDEA. This research was carried out under grant number 2048840 from the National Science Foundation. This is PMEL contribution 5766.

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
