# Peer review of "Atmospheric and Ocean CO2 Measurements in the South Indian Ocean Made by Two Uncrewed Surface Vehicles in 2022 and 2023"

_Earth System Science Data, 2025_

## Author Comment (AC1)

We would like to thank **Reviewer # 1** for their very thorough review of our manuscript and thoughtful suggestions. In particular, we appreciate noting the references listed that are not used in the paper – that was an oversight on our part, and we should have been more careful in reviewing that. We have also added the suggested updated references in appropriate places, unless otherwise noted in the response to major comments.

Below, we respond to the more significant suggestions, in the order they appear. The reviewer's comment is in italics, while our **Response** follows.

1. *"However, they do not detail the accuracy/precision of the data and this should be added (e.g. in Table 1)."*

**Response**: Unfortunately, the authors do not have this information for the standard sensors on the USVs, or for the radiometers which were supplied by the SO-CHIC mission. We have asked Saildrone if they have this information and they sent us to the manufacturer sites, where we have searched and cannot find that information readily available. We note that we already list the manufacturer and model numbers in Table 1. Since these are standard off-the-shelf instruments, we don't feel this level of detail is necessary and note that it is rarely provided in other manuscripts we have seen. We already noted the accuracy of the CO2 sensor in the text (line 53 and line 176).

We also noted on line 201 of the original manuscript that standard deviations of the 11-sec burst are included (where available). This can be used to estimate precision. We have added a statement to the original sentence to make this clear:

"Where available, the standard deviation of the 11-second burst measurement is also included, which can be used as an estimate of the precision of the measurement. The standard deviation of the minute-sampled data can also be used as a measure of the accuracy of the 1-hour averages."

If the editor feels that manufacturer precision numbers are required, we will do our best to gather these. However, we feel this response should be sufficient.

2. *"… not discussed or highlighted in the MS, is that pCO2 was lower than in the atmosphere for almost all periods (Figure 4), i.e. a CO2 sink; this is different than observed by Sutton et al (2021) using data from the first circumpolar USV and who observed CO2 sources in winter."*

**Response**: Thanks for pointing this out. We felt this was obvious enough to not mention, but we have added a statement on this in the revised manuscript around lines 345-352

"The ocean $pCO_2$ along the SD1039 track (Fig. 4a,c) primarily varies between $360 - 385$ µatm, except for short excursions where the water $pCO_2$ abruptly drops or rises by up to 20 µatm for a period lasting less than a day (in most cases, only a few hours). Generally, the ocean $pCO_2$ is lower than the atmospheric $pCO_2$, indicating the ocean was acting as a sink for $CO_2$ during the USV transits. Near the end of the SD1038 transit (south of 45°S), ocean $pCO_2$ increases to be close to that of the atmospheric $pCO_2$. Some spikes are larger than +40 µatm near the end of the record of SD1039, indicating a short-term higher concentration of $pCO_2$ in the surface waters

than in the atmosphere. This suggests the potential for outgassing of $CO_2$ from the ocean to the atmosphere during these periods, such as was observed in the 2019 Saildrone mission (Sutton et al., 2021), but more work would be required to fully quantify this."

3. *"I was wondering why authors did not compare their data with the 2019 SD (Sutton et al, 2021). Is it because the tracks were not in the same region? .... Note also that fCO2 data from several cruises are available in the investigated region in 2022-2024. Although the periods are not exactly the same (i.e. no direct cross-over), comparisons might highlight what the USV offers that could be not interpreted from shipboard data."*

**Response**: We took a quick look at this, and while there are some SOCAT tracks that occur near the location of our USVs, they make measurements months to over a year apart. The only track that occurs during the approximate time of our mission is some 1000km to the south. The 2019 Saildrone mission had a transit significantly further south than our mission, was 4 years earlier, and crossed the Indian Ocean at a very different time of the year (May to June). Such comparisons would be comparing apples to oranges, would provide no relevant information as to the accuracy of our observations due to space and time differences, and is not pertinent to a paper describing the mission and dataset.

We note that the purpose of ESSD is to publish "articles on original research data (sets)," and not to publish new scientific research." In fact, the first two authors of this paper recently had a paper rejected from ESSD because it was deemed to have too much new scientific analysis in addition to describing a dataset. We were asked to submit to a different journal because of this.

We have added a new opening paragraph at the start of Section 3 to explain our arguments:

"Here we only analyze observations from the SOS mission and present a preliminary analysis of observed variations that may correlate with the eddy-type. Observations cannot be directly compared to other measurements made by ships, drifters, or other USVs because no other observations were made within 500 km or within 3 months of the two USV transits (based on the current SOCAT database). While one goal of the mission was to attempt a crossover with a biogeochemical Argo float (as done with the 2019 Saildrone mission (Sutton et al., 2021), this was not possible during the SOS mission. Any observations made near the same area were collected perhaps a year earlier or later. The 2019 Saildrone mission, for example, occurred 3-4 years earlier and crossed the South Indian Ocean farther south than SD1039. Any comparison would have to account for spatial and temporal differences and is best suited for a scientific investigation which is beyond the scope of this data description paper."

4. *"... as the USV also recorded skin temperature, it might be useful to present these data along with SST to evaluate their differences and discuss potential bias on the air-sea CO2 fluxes estimates (e.g. Ford et al, 2024). Is there a link of the SST-Tskin in specific regions (e.g. in eddies) or this has no impact on CO2 fluxes in the investigated region."*

**Response**: Thanks for the suggestion. We do not compute fluxes in this paper (judging that to be more of a scientific investigations). Additionally, the amount of SST-skin temperature is limited

to less than a month, and it is intermittent at that. This is because we had to limit which instruments were turned on and the priority was given to the ASCVO2 system, the seawater property system, and the ADCP. Upon re-reading the data section, we realized this was not made clear, so we add some text in Section 2 (around lines 159-162) of the revised text:

"However, it must be noted that due to ongoing power issues, only the seawater properties, the ASVCO2® system, and the ADCP system were operated nearly continuously with minimal outages. Most other parameters have significant gaps due to instruments being turned off to conserve power, as they were deemed of lesser importance compared to maintaining the USV power and primary instruments. We have archived all available observations in the databases (Table 2), but users should not expect complete records of many variables other than the seawater properties and chemistry data."

Because of the limited skin temperature available, we do not feel such a comparison would be useful.

*5. "Line 41-42: Authors refer to Fig. 3 from Bakker et al., (2016) who present data for 1957-2014. Since 2014, millions of new data have been included in SOCAT and authors may show maps (e.g. 4 seasons) of the most recent SOCAT version (v2025, Bakker et al, 2025) and inform that much less data were obtained in austral winter as noticed in line 42. In such maps the tracks of the UVS would be highlighted (suggestion)."*

**Response**: Thanks for the suggestion, but as the reviewer notes, nothing has significantly changed in terms of data availability during winter in this region since Bakker et al. (2016). We feel this statement is sufficient for the original motivation of the mission. We also want the first figure in the paper to be focused on the mission. Therefore, we respectively decline to add this figure.

6. *"downstream of the Kerguelen Plateau". For reader not familiar with this region, please add on the map (figure 1) the names of some locations (Kerguelen, Kerguelen Plateau, Crozet etc…)" and "Lines 77 and 101: Figure 1: The track of SD1039 seems to be located in the SAZ between the STF and the SAF. I would suggest add the location of the STF (and AGF) on the map. Also, I think the PF should be south of Kerguelen."*

**Response**: We have added the Subtropical Front, bathymetry contours, the Kerguelen Plateau (KP) and Crozet Island (a white X) to Figure 1. We decided not to include Kerguelen Island (because that is not mentioned in the manuscript). We did not add the AGF because it made the figure busy in that area and we wanted to highlight the track of SD1038. The PF may meander slightly south of Kerguelen Island, but it is generally north of the broad Kerguelen plateau. The SACCF is to the south of KP.

7. *"C-09: Line 174: In addition to NCEI, authors may also indicate that fCO2 data for one UVS are available in SOCAT-v2025 (Bakker et al, 2025)."*

Thanks for pointing that out. We have added that comment, as well as that SD1038 data will be added in time for the next release, around Table 2:

"For users familiar with the Surface Ocean CO2 Atlas (SOCAT, https://socat.info), the chemistry data for SD1039 has been uploaded and is currently available in the current database (SOCAT-v2025, Bakker et al, 2025). The data for SD1038 had not been processed in time for inclusion in the current atlas but will be added in the next one."

8. *C-10, Line 254: Authors write: "The most obvious signal in measured pCO2 by SD1038 during its June-July transit from South Africa is an increase in pCO2 values from ~350 µatm at 35° S to ~405 µatm at 50°S as the vehicle moved southwards (Fig. 4a and 4b). These values are within an expected range, as Shadwick et al. (2023) documented seasonal variations at a similar latitude in a mooring south of Tasmania with a peak (380-400 µatm) around July/August".  As the SD1038 was in the western Indian sector, maybe compare with other results in this region (e.g. figure 3 in Metzl et al, 2006 for summer and winter) not only south of Tasmania. It might be also useful to compare the SD data with the climatology from Takahashi et al (2009) or from Fay et al (2024), especially because the SD1038 data were not used to compose the climatology.*

**Response**: We already compare our results to a recent climatology (Landschützer et al., 2020) that is arguably more up to date than the older climatology suggested. The Fay et al. climatology is for CO2 flux, not pCO2. Since we do not discuss CO2 flux in this manuscript (believing that is more of a scientific use of these observations), we do not feel it is relevant here. We selected the Shadwick et al paper because it is more recent and closer in time to our observations compared to the 2006 Metzl study. We also show the general latitude/longitude changes in pC02 we observe are consistent with the expected climatology values (except for a bias – more on that in the next response).

10. C-19: *Figures 4: Atmospheric pCO2 presented in the plots range between 400-420 (in µatm). Recall in the text that pCO2atm here is at atmospheric pressure, i.e. xCO2 in ppm would be about 412-416 ppm in the southern hemisphere in 2022-2023. By the way, a comparison with atmospheric CO2 recorded at Stations Crozet would also validate the atmospheric pCO2 data from the USV.*
**Response**: Thanks for pointing this out. We have added a few sentences discussing the observed atmospheric pCO2 and comparing to the Crozet station CO2 observations, near lines 291-298 of the revised manuscript:

"Atmospheric $pCO_2$ has a mean value of 410 µatm with a standard deviation of 3.5 µatm for both USVs. Observations of atmospheric $CO_2$ made at Crozet Island in the Indian Ocean (46.4°S, 51.8°E) between Sept. 4, 2022 and April 26, 2023 has a mean of 415.2 ppm (standard deviation = 0.46). Converting to $pC0_2$ in µatm using average and fixed air pressure (1 atm) and water vapor pressure (0.015 atm), this corresponds to approximately 409 µatm. Recalling that the accuracy of $pC02$ measurements from the ASVCO2 system is ± 2 µatm, the USV measurements are consistent with observed $pC02$ in the region. The Crozet CO2 measurements were downloaded from the NOAA Global Monitoring Laboratory (https://gml.noaa.gov/data/dataset.php?item=crz-co2-flask; Lan et al., 2025) on 30 Jul 2025.

9. *Comments on Figure 5 and relevant discussion. C-11, Line 260: Authors notice a bias between SD data and the climatology from Landschutzer et al (Figure 5). What was the reference year for this climatology ? Would the bias explained by the increase of oceanic pCO2? Also, reviewer asks to show difference to better highlight the bias.*

**Response**: We did state the time frame of the climatology as 1988 – 2015 (corrected now) in the figure caption. However, there is no statement in the Landschutzer et al paper explicitly stating the reference date used (either in older papers, newer papers, or the dataset itself), but after emailing him, he told us it was 2006-2007. We have added that to the caption (and also corrected seasonal to monthly):

"Figure 5: Observed ocean $pCO_2$ along the transects of SD1038 (a) and SD1039 (b). Also shown are values from a mean monthly climatology (red, data from Landschützer et al., 2020a,b). The climatology is based on all available data from 1988 to 2015 and so will reflect a mean state in 2006-2007 (P. Landschützer, personal communication)."

The reviewer makes a good point about the potential explanation of the bias, and we should have thought of this. In fact, assuming ocean pCO2 increases at the same rate as atmospheric pCO2 measured at Crozet ($\sim$ 2 µatm/year since 2005) then, the expected climate-induced change between January 2007 and December 2022 would be 32 µatm.

We have revised the paragraph to reflect this analysis (lines 300-320 in the revised manuscript).

"The most obvious signal in measured oceanic $pCO_2$ by SD1038 during its June-July transit from South Africa is an increase in $pCO_2$ values from $\sim$350 µatm at 35° S to $\sim$405 µatm at 50° S as the vehicle moved southwards (Fig. 4a and 4b). These values are within an expected range, as Shadwick et al. (2023) documented seasonal variations at a similar latitude in a mooring south of Tasmania with a peak (380-400 µatm) around July/August. Comparing the $pCO_2$ to a mean monthly climatology (Fig. 5) we confirm the shift in values of oceanic $pCO_2$ in SD1038 around 45°S is consistent with the mean state in that region for the time of year. We do note a bias between the measurements of both SD1038 and SD1039 and the climatology (Fig. 5) of approximately 25-30 µatm (SD1038/1039 higher). The climatology was based on an average of observations from 1998 to 2015. Although the exact epoch represented by the climatology is not explicitly given in the reference paper or dataset (Landschützer et al., 2020a,b), P. Landschützer confirmed to us via email that the mean epoch is 2006-2007. The mean rate of change in atmospheric $pCO_2$ at Crozet since 2005 is $\sim$ 2 µatm $yr^{-1}$ (based on the approximate pressure values stated previously). Multiplying this rate by the time difference (15-16 years) between our observations and the climatology epoch gives a climate-induced change of 30 to 32 µatm. Since oceanic $pCO_2$ should follow trends in atmospheric $pCO_2$ assuming an equilibrium state (e.g., Fay et al., 2024), one would expect measurements of oceanic $pCO_2$ to have changed by this much on average. This is approximately the bias we observe, so we conclude the observed bias with the climatology is primarily due to increasing $CO_2$ concentrations since 2006-2007 and any smaller deviations are interannual fluctuations and using direct pressure/temperature observations instead of climatological means."

Finally, we don't feel including the difference to the plot would add anything. The bias is obvious in the current plot, and adding a third curve would make the figure busy. Moreover, the current figure is useful for seeing latitude/longitude shifts in $pCO_2$ that are common in both the USV data and the climatology. We feel the added discussion should be sufficient to address the reviewer's concerns.

10. *C-14: Line 285: Could you recall how the Chl-a in µg/l was derived from fluorescence data.*

**Response**: This was either done by Saildrone, Inc. (or by the instrument), so we do not have this information.

11. *C-15: Line 286: To separate the effect of biology it might be useful to show the SST in Figure 6 and pCO2 normalized to SST at 13 or 15°C.*

And

*Line 309: Is the largest pCO2 change linked to eddy or meander in the frontal structure. Like for the SST (comment above), it would be interesting to show the sea surface salinity records (rapid change of salinity between 35.5 and 34 at the SAF is common in this region).*

**Response**: We have added the SST and SSS data in two new plots similar to the Figure 4 (Figures 7 and 8 in revised manuscript). Also included are the frontal positions.

We believe going too much into the analysis of whether the pCO2 anomaly that correlates with Chl is driven by biology or changes in SST is beyond the scope of this data description paper. However, we do add some commentary on the relative SST changes observed relative to the Chl observations, and pCO2 changes due to SST (lines 361 to 361 in the new manuscript):

"We note that the SST dropped by nearly 5°C during this period (Figure 7), suggesting increased upwelling during this period. This is consistent with increased nutrient availability for a phytoplankton bloom. The temperature drop is unlikely to be the primary reason for the $pCO_2$ change, as a shift in SST from ~16°C (just before the drop) to 11°C in the period of low $pCO_2$ (Figure 7) will only cause a change of change of ~ 1 µatm, which is 10% of the observed change (Figure 6)."

We also add a small paragraph noting the changes in SST and SSS associated with fronts. On lines 386 – 393.

"The surface salinity (SSS) is also plotted (Figure 8). It is apparent from both Figure 7 and 8 the changes in SST and SSS as the two USVs cross fronts. This is particularly noticeable in the measurements of SD1038 as it crosses the STF, the SAF, and the PF. It is seen to a lesser extent when SD1039 moves into the Subantarctic Zone between the STF and SAF in December 2022. Note that the maximum gradients in SST and SSS do not perfectly align with the mean frontal positions of Orsi et al (1995). Fronts are highly variable in time and space, and mean frontal positions are only an approximation. For example, Kosempa and Chambers (2014) noted that a

frontal calculation based on upper ocean zonal transport placed the STF south of Africa substantially further north than in Orsi et al. (1995), which is also observed by SD1038 in SST (Figure 7) and SSS (Figure 8)."

12. *Figures 2: Add on the map the Eddy number as listed in Figure 7. Would Figure 2 moved juste before figure 7*

**Response**: Moving Figure 2 just before Figure 7 would not work very well because of the relevant discussion with each. Also, because of the non-constant speed of the USV, it would be impossible to align the two plots perfectly (see, for example, Figure 4 and the location (panel b) vs time (panel c) of the minimum pCO2.

We have added eddy numbers to Figure 2, however.

---

## Author Comment (AC2)

We would like to thank **Reviewer # 2** for their positive review of our manuscript and thoughtful suggestions.

Below, we respond to the more significant suggestions, in the order they appear. The reviewer's comment is in italics, while our **Response** follows.

1. *"L177-178: Data from the DuraFET pH sensor were not reported in the dataset (all values for pH are -999 in the files) because the sensor was uncalibrated. However, the manuscript mentions that this data was used for the flagging of CO2 measurements. To my knowledge the procedure to assess quality flags for CO2 measurements from the pH data are not described in Sutton et al. 2024 and Sabine et al. 2020. Some information needs to be given on how the pH values were used to do quality checking."*

**Response**: That section has been revised to add more detail:

"While the $ASVCO_2$ sensor package also included a DuraFET pH sensor, these data are not included in the files as they are uncalibrated. They were only used (along with internal $CO_2$ system diagnostics) to quality check and flag of $CO_2$ measurements. This was done by calculating covariance of pH and $CO_2$ observations over segments where there were significant $CO_2$ deviations. Existence (or lack) of covariance between $CO_2$ and pH outliers was used as independent evidence that $CO_2$ data were good (or questionable)."

2. *"L210 : Please give a reference to access the " Mesoscale Eddy Trajectory Atlas"*

**Response**: That location was placed in the acknowledgements because it is a very long website (and has no DOI that we are aware of). However, we have added an explicit reference in the revision (META3.2, 2025), then add a new reference in the reference list (assuming this is allowed by the Journal, since it does not have a DOI. This is a standard product for users interested in tracking mesoscale eddies and putting the location in the acknowledgements is standard practice.

New reference:

META3.2: Mesoscale Eddy Trajectories. SSALTO/DUACS/AVISO+. https://www.aviso.altimetry.fr/en/data/products/value-added-products/global-mesoscale-eddy-trajectory-product/meta3-2-exp-nrt.html), downloaded January 4, 2025.

3. *"L216 to 227: The methodology used to assess if the Saildrone is outside or inside an eddy is described in this section. I believe that it would be useful to the reader to also indicate that the variable EDDY_DIRECTION has only three values : 1 for Anticyclonic Eddy, 0 for no Eddy and -1 for Cyclonic Eddy."*

**Response**: Done. New text reads:

"Because the goal of the SOS mission was to measure pCO2 within different eddies, we have also provided an estimate of whether the USV was in an eddy or not, along with the type of eddy (cyclonic, anticyclonic) in the main mission datafile. The variable EDDY_DIRECTION in the file has three values (1 for anticyclonic, -1 for anticyclonic, and 0 for not within an eddy)."

4. *"L260-265: The authors mention a bias between the pCO2 values from the saildrone and the mean monthly climatology. It should be noticed that the mean monthly value of this climatology is based on the mean for a given month of the values spanning from 1988 to 2020 whereas the measured values are from 2022-2023. The temporal increase of the atmospheric CO2 fraction could (at least partially explain this bias)."*

**Response**: Reviewer # 1 also pointed this out and asked for a more thorough analysis, by looking at changes in the atmospheric CO2 measured at the Crozet station in the Indian Ocean. We have first have added a comparison of the observed atmospheric pCO2 and that made at Crozet around lines 306-312 of the revised manuscript:

"Atmospheric $pCO_2$ has a mean value of 410 µatm with a standard deviation of 3.5 µatm for both USVs. Observations of atmospheric $CO_2$ made at Crozet Island in the Indian Ocean (46.4°S, 51.8°E) between Sept. 4, 2022 and April 26, 2023 has a mean of 415.2 ppm (standard deviation = 0.46). Converting to $pC0_2$ in µatm using average and fixed air pressure (1 atm) and water vapor pressure (0.015 atm), this corresponds to approximately 409 µatm. Recalling that the accuracy of $pC02$ measurements from the ASVCO2 system is ± 2 µatm, the USV measurements are consistent with observed $pC02$ in the region. The Crozet CO2 measurements were downloaded from the NOAA Global Monitoring Laboratory (https://gml.noaa.gov/data/dataset.php?item=crz-co2-flask; Lan et al., 2025) on 30 Jul 2025.

In the following paragraph, we discuss the bias and explain it is primarily due to change in time from the mean of the climatology. Note that the climatology documentation never explicitly states the reference time, but we reached out to P. Landschützer and he provide it:

"... We do note a bias between the measurements of both SD1038 and SD1039 and the climatology (Fig. 5) of approximately 25-30 µatm (SD1038/1039 higher). The climatology was based on an average of observations from 1998 to 2015. Although the exact epoch represented by the climatology is not explicitly given in the reference paper or dataset (Landschützer et al., 2020a,b), P. Landschützer confirmed to us via email that the mean epoch is 2006-2007. The mean rate of change in atmospheric $pCO_2$ at Crozet since 2005 is ~ 2 µatm yr$^{-1}$ (based on the approximate pressure values stated previously). Multiplying this rate by the time difference (15-16 years) between our observations and the climatology epoch gives a climate-induced change of 30 to 32 µatm. Since oceanic $pCO_2$ should follow trends in atmospheric $pCO_2$ assuming an equilibrium state (e.g., Fay et al., 2024), one would expect measurements of oceanic $pCO_2$ to have changed by this much on average. This is approximately the bias we observe, so we conclude the observed bias with the climatology is primarily due to increasing $CO_2$ concentrations since 2006-2007 and any

smaller deviations are interannual fluctuations and using direct pressure/temperature observations instead of climatological means."

5. *L251 : I would suggest renaming this section. The current title for this section "Analysis of the Chemistry data" is maybe confusing. I believe that the authors only make a preliminary illustration of the potential of the data rather than a real analysis of the data."*

**Response**: We have renamed the Section to "Discussion of Observations." Note that we have added figures of SST and SSS as well and added some brief discussion of changes of those parameters as the USVs move across fronts, at the Request of Reviewer # 1.

6. "*Figure 1 : This figure could certainly be improved. It is sometimes difficult to distinguish the trajectory of the saildrone due to the choice of the colors. The legend of the figure should also mention white more detail what type of data have been used for the standard deviation of the surface height variability."*

**Response**: We recognize this is a busy figure. And Reviewer # 1 asked us to add even for information (the STF, location of Crozet, the Kerguelen Plateau, and bathymetry). We have adjusted the size of the dots and diamonds indicating the USV tracks and changed the colors to orange and dark blue to provide more contrast. We hope this is sufficient. Otherwise, we feel like we would need to create multiple figures to convey the information. We also note that the USV track information is also shown in other figures (4, 7, and 8) when we show pCO2, SST, and SSS.

7. "*Figure 2 and figure 3 : The color of the dots used to describe the center of the eddies is easy to distinguish. Maybe, it is not useful to give this information to these figures."*

We feel the color of the central dot is not that distracting so choose to keep it.

8. *Several typos/minor corrections.*

**Response**: All have been made. We found several more incorrect subscripts in the revision review. "Ocean color" was removed and we now state "Chlorophyll α based on fluorescence" throughout. This also occurred in the abstract.

---

## Author Response (AR2)

**Dear Dr. Gazeau,**

Thank you for your quick review of our revised manuscript. Per your request, we have added a new column to Table 1, listing precision/accuracy for the measurements as best we can describe them. Most are based on the standard deviation of sub-second burst measurements (where available) – these are also provided in the full data record as noted previously in the text. A few are from the manufacturer specifications or literature.

We were unable to find accuracy values for the CCMP gridded winds – these are not explicitly provided as a value in the literature. In that case, we direct the reader to the appropriate paper so they can use their best judgement. The only measurement where we do not provide a precision/accuracy value is for the outgoing longwave radiation measurement. This was not an instrument for our mission, but an instrument provided by the SO-CHIC mission as mentioned in the text. We can find absolutely no mention of accuracy or precision estimates in the literature or manufacturer on this instrument, merely algorithms for converting the actual measurements of microwave irradiance into a flux, which was done onboard. Therefore, we have stated "Unknown. Deployed by SO-CHIC mission" for this one.

We feel this is the best we can do for this, and we note again that we find few papers in the literature using similar instruments that go into this level of detail. Hopefully, this is now acceptable.

We have also modified Table 2 to refer to the specific variables stored in the data set in Table 1. Trying to list ALL variables in this table would be cumbersome. Again, we hope this is now acceptable.

Please let us know if you have any further requests.

Regards,

Don Chambers (for the author team)